# An Integrative Neuro-Psychotherapy Treatment to Foster the Adjustment in Acquired Brain Injury Patients—A Randomized Controlled Study

**DOI:** 10.3390/jcm9061684

**Published:** 2020-06-02

**Authors:** Antoine Urech, Tobias Krieger, Eveline Frischknecht, Franziska Stalder-Lüthy, Martin grosse Holtforth, René Martin Müri, Hansjörg Znoj, Helene Hofer

**Affiliations:** 1Department of Neurology, Inselspital Bern, Bern University Hospital, 3010 Bern, Switzerland; rene.mueri@insel.ch; 2Department of Psychology, University of Bern, 3012 Bern, Switzerland; tobias.krieger@psy.unibe.ch (T.K.); eveline.frischknecht@ptp.unibe.ch (E.F.); martin.grosse@psy.unibe.ch (M.g.H.); hansjoerg.znoj@psy.unibe.ch (H.Z.); 3Department of Neurology, Spitalzentrum Biel, 2501 Biel, Switzerland; Franziska.Stalder@szb-chb.ch; 4Department of Psychosomatic Medicine, Bern University Hospital, 3010 Bern, Switzerland; 5Gerontechnology and Rehabilitation Group, ARTORG Center for Biomedical Engineering Research, University of Bern, 3008 Bern, Switzerland

**Keywords:** acquired brain injury, cognitive rehabilitation, neuropsychology, psychotherapy, adjustment disorders, stroke

## Abstract

Adjustment disorders (AjD) with depressive symptoms following an acquired brain injury (ABI) is a common phenomenon. Although brain injuries are increasing more and more, research on psychological therapies is comparably scarce. The present study compared, by means of a randomized controlled trial (RCT), a newly developed integrative treatment (Standard PLUS) to a standard neuropsychological treatment (Standard). Primary outcomes were depressive symptoms assessed with the Beck Depression Inventory (BDI-II) at post-treatment and 6-month follow-up assessment. In total, 25 patients (80% after a stroke) were randomized to one of the two conditions. Intention-to-treat analyses showed that the two groups did not significantly differ either at post-treatment nor at follow-up assessment regarding depressive symptoms. Both treatments showed large within-group effect sizes on depressive symptoms. Regarding secondary outcomes, patients in the Standard PLUS condition reported more emotion regulation skills at post-assessment than in the control condition. However, this difference was not present anymore at follow-up assessment. Both treatments showed medium to large within-group effects sizes on most measures for patients suffering from an AjD after ABI. More research with larger samples is needed to investigate who profits from which intervention.

## 1. Introduction

A stroke is one of the most common causes of disability acquired in adulthood and is a common cause of death and adult disability worldwide [1]. In the last 20 years, the incidence rate of ischemic and hemorrhagic stroke increased significantly [2]. Improvements in medical treatment for stroke has advanced rapidly, and more people survive but live with the consequences of stroke [3].

Patients with stroke have to cope with direct consequences of the brain injury (e.g., memory or attention impairments, language problems, hemiparesis, and visual problems) as well as secondary consequences (e.g., loss of experiences at work and in social functioning). These fundamental and lifelong changes trigger a sense of substantial uncertainty in most patients, as essential areas of their lives are threatened [4,5]. As a consequence, it is not surprising that around 50% of acute stroke survivors have residual major physical or cognitive deficits and need assistance for adaptation and coping with the complex physical and social sequelae of stroke [6].

People with acquired brain injury (ABI) are at increased risk of developing emotional disturbances compared to the general population [7]. Many stroke patients adjust well to their changed life conditions, whereas others develop serious symptoms of mood disorders such as feelings of worthlessness and sense of loss, and complete restoration is not always possible [4]. Studies estimate that about one-third of patients suffer from mood disorders after stroke [8,9,10]. It is generally assumed that emotional disturbance following ABI might be a direct result of structural brain lesions (e.g., the location of the lesion) [11,12]. Alternatively, it is suggested that physical limitation (e.g., hemiparesis), cognitive impairments (e.g., loss of memory or language), and social and psychological stressors (e.g., the ability to cope with the illness) are associated with mental disorders [13,14,15]. Thus, it seems reasonable to assume that emotional disorders following stroke appear to be a biopsychosocial multifactorial illness [16,17,18,19].

Various studies have shown that untreated adjustment disorders (AjD) can severely restrict the potential for rehabilitation of stroke patients and can hinder their re-entry into work. Moreover, a Danish study showed that around 50% of these patients had left their jobs permanently or were still on sick leave one year after stroke [20]. Another study revealed that six years after stroke, 84% have returned to work but only 35% reached the same level of work ability [21]. In the long term, they also represent the greatest strain on the family [22]. Nevertheless, so far, surprisingly little attention has been given to AjD in terms of research, despite the increased number of AjD in primary care [23,24]. One reason might be that neither the international classifications system DSM-IV nor the international classification of diseases and related health problems ICD-10 includes specific diagnostic criteria for mental disorders such as AjD due to brain injury. Hence, it is treated differently in various studies. Moreover, some researchers define “mental disorder after stroke” as major or minor depression [25,26], while others classified it as an AjD brought on by a stressful life change [27]. According to the DSM-IV, AjD is characterized by persistent maladaptive emotional or behavioral response to an identifiable psychosocial stressor, to which patients experience difficulties adjusting after a stressful event [28]. Numerous studies have shown that untreated mood disorders or AjD have a predictive negative impact on daily activities [29], cognition [30], and rehabilitation [31,32], as well as increased mortality rate [33]. Moreover, untreated psychological problems can further exacerbate social difficulties after ABI, often leading to significantly reduced quality of life, substantially reduced psychological coping capacities, and additional deterioration [34].

To date, many theoretical considerations about ways of dealing with mood disorders after ABI exist in order to improve rehabilitation outcome [35,36,37]. In general, management of mood disorders or AjD after stroke includes antidepressants and psychotherapy [10]. Regarding antidepressants, a Cochrane review [38] including 56 randomized controlled trials provided strong support for the efficacy of antidepressants (e.g., selective serotonin reuptake inhibitors; SSRIs) following stroke in patients with mood disorders. Despite the positive effects of antidepressant treatment, there is no conclusive evidence regarding the optimal length of antidepressant treatment for mood disorders [39]. Moreover, antidepressant treatments also may have many side effects and risks, including an increased risk of hemorrhagic complications and falls [40].

Regarding psychotherapy, a systematic review found preliminary evidence supporting the efficacy of cognitive behavior therapy (CBT) for depression following ABI [41]. Another meta-analysis revealed an overall effect size of 0.69, suggesting a medium effectiveness of psychological interventions on mood disorders compared to control conditions [42]. Therefore, CBT interventions for psychological distress in people with ABI might have the potential to improve psychological well-being, but there is still a lack of evidence-based treatment studies for mental disorders following stroke [5]. A recent Cochrane review by Gertler and colleagues [43] identified three eligible studies evaluating CBT or mindfulness-based therapy compared to a control intervention (waitlist group or supportive therapy) for ABI patients suffering from mood disorders. In terms of reduction of depressive symptoms, they did not find an effect in favor of treatment, but there was very low-quality evidence (e.g., high attrition bias), small effect sizes, and wide variability of results. Moreover, Ashman and colleagues [44] evaluated in a randomized controlled trial comparing CBT and supportive psychotherapy for the treatment of depression following traumatic brain injury. The main results revealed that there was no significant difference between groups regarding the level of depressive symptoms at post-assessment. Furthermore, a recent published pilot randomized controlled trial showed that acceptance and commitment therapy (ACT) may facilitate adjustment after traumatic brain injury and revealed significant changes regarding stress and depressive symptoms after post-assessment compared to an active control condition. However, this reduction of stress and depressive symptoms was not maintained at follow-up assessment nor did the primary outcome (psychological flexibility) differ significantly at any assessment point between the two groups [45]. In a non-controlled pilot study, Hofer and colleagues [46] tested an integrated treatment program in which neuropsychological interventions are supplemented by psychotherapeutic interventions to foster the adjustment process. Results indicated a large within-group effect size of 1.30 on the Beck Depression Inventory (BDI-II). In addition, patients showed significant within-group changes regarding three coping styles (decrease of rumination, increase of searching for social support, and cognitive coping).

The combination of neuropsychological and cognitive–behavioral approaches for treating psychological sequelae following ABI has drawn more attention during recent years. Unfortunately, a paucity of randomized controlled trials limits drawing convincing conclusions [47]. A recent systematic review showed that CBT and family and systemic therapies are recommended at all stages in the evolution of appropriate therapeutic interventions of mood disorders following ABI [48].

The main aim of the present study was to investigate by means of a randomized controlled trial whether an integration of neuropsychological and psychotherapeutic treatment is more effective than a neuropsychological treatment alone in ABI patients diagnosed with an AjD, which is one of the most under-researched psychiatric disorders [49]. Furthermore, we hypothesized that the intervention group (Standard PLUS) will show a greater reduction in depressive symptoms (BDI-II) compared to the control condition (Standard group, i.e., neuropsychological treatment alone).

## 2. Materials and Methods

This randomized controlled trial (RCT) compared an intervention group with an active control group. The trial was registered with http://www.clinicaltrials.gov/ (NCT01039857) and was approved by the Ethics Committee of the Canton of Bern, Switzerland (012/08) on 23/10/09.

### 2.1. Patients

Patients were recruited via the outpatient setting of the University Neurorehabilitation of the Department of Neurology, University Hospital, Inselspital, Bern. Inclusion criteria were (a) being between 18 and 66 years, (b) time passed since stroke having been more than six months (e.g., with regard to spontaneous remission [50]), (c) having a diagnosis of an adjustment disorder (DSM-IV: 309.x, acute or chronic), and (d) having the sufficient ability to communicate in German. Criteria for exclusion were (a) the presence of another chronic disease (e.g., multiple sclerosis, sarcoidosis, Parkinson disorder, neurodegenerative illness, chronic pain disorder, rheumatic disorder), (b) a prior history of a neurological disease, (c) a prior history of a mental disorder, and (d) acute suicidality or violent behavior. 

After returning the signed informed consent form, participants completed an initial neuropsychological assessment and were interviewed by the therapist using the Structured Clinical Interview for DSM-IV Axis I Disorder (SCID-I) [51] to check the diagnostic criteria for an adjustment disorder. All interviews were supervised by the senior author. After the interview, participants were asked to complete the baseline questionnaires. In addition, a comprehensive neuropsychological examination, including measures of memory, attention, and executive and visual functioning was performed for each patient before the beginning of the neuro-psychotherapy to ensure insufficient cognitive abilities. Patients with severe cognitive impairments or sufficient language abilities were excluded from the study.

At the beginning of recruitment, we mainly focused on stroke patients. Due to problems with recruitment of patients, (e.g., challenging medical condition in the context of the given health care system and homogeneity of the study sample), we widened the inclusion criteria during the trial, and any kind of non-progredient ABI (*n* = 5; e.g., traumatic brain injury or encephalitis) was included. A total of 25 participants who fulfilled inclusion criteria were randomly assigned to one of the two conditions. Randomization took place at the Insitute of Social and Preventive Medicine University Bern, which gave feedback to the researcher regarding group allocation according to a pre-generated randomization sequence. Participants were randomized in a 1:1 ratio. The flowchart of the present study is depicted in Figure 1.

### 2.2. Measures

The following measures were applied at several time points (baseline, post-treatment and 6-month follow-up). The questionnaires were handed over to the patients personally and administered on paper. In addition, Goal Attainment Scaling (GAS-R) was performed at pre- and post-treatment (see below). 

### 2.3. Primary Outcome

The primary endpoint of the present study was the Beck Depression Inventory II (BDI-II; [52]) at post-treatment and 6-month follow-up after post-treatment. The BDI-II is a self-report measure consisting of 21 items describing symptoms of depression. The total score ranges from 0 to 63, with a higher score indicating more depressive symptoms. The BDI-II has been widely used in different populations, including patients with a brain injury [53]. The internal consistency in the present sample was α = 0.90.

### 2.4. Secondary Outcomes

Quality of Life. To assess the quality of life, a short form of the World Health Organization Quality of Life (WHOQOL-BREF) was administered [54]. The WHOQoL-BREF uses 26 items to measure self-assessed quality of life with regard to mental well-being, physical well-being, social relationships, and the environment. In the present study, we report a sum score over all items. Cronbach’s α in the present study was 0.87.

#### 2.4.1. Acceptance of Disability

To assess the level of acceptance of disability, a revised version of the Acceptance of Disability Scale (ADS) was applied [55,56]. The ADS has 32 items and uses a four-point Likert scale which measures an individual’s overall acceptance of a disability (e.g., “With my disability all areas of my life are affected in some major way”). Cronbach’s α in the present study was 0.81.

#### 2.4.2. Awareness

In order to assess the level of awareness, we used the Awareness Questionnaire (AQ) [57]. The AQ was developed as a measure of impaired self-awareness after traumatic brain injury (TBI). The AQ includes three forms; one form is completed by the person with TBI (AQ_P), one by a significant other of the patient (AQ_R), and one by the therapist (AQ_T). The self-rated and family/significant others forms include 17 items (e.g., “How well can you concentrate now as compared to before your injury?”), while the therapist form has 18 items (e.g., “How well can the patient concentrate now as compared to before his/her injury”). On each form, the abilities of the person with TBI to perform various tasks after the injury as compared to those before the injury and rated on a 5-point Likert scale ranging from 1 (“much worse”) to 5 (“much better”) [58]. Cronbach’s α in the present study for the self-rated form was 0.81, for the family/significant others form, 0.83, and for the therapist form, 0.61.

#### 2.4.3. Illness Coping

Different coping behaviors of the patients were assessed with the Trier Illness Coping Scales (TSK) [59]. The standardized questionnaire includes 37 items with a 6-point Likert scale, which are combined into five subscales, and characteristics of cognitive and behavioral strategies of coping are assessed. The five subscales are the following. *Rumination* specifies brooding thoughts about previous illness-related problems. Patients with high scores on this scale are searching for causes of their illness in the past and draw comparisons to the time before disease onset (e.g., “I was worried whether the doctor may help me”). *Defense of threat* is an intrapsychic coping style that combines palliative cognitive reactions like revaluation and downward comparison (e.g., “I had a good time with other people”). *Search for social integration* describes turning toward the social environment to mobilize emotional support and to distract from illness-related problems (e.g., “I enjoyed myself outdoors in nature”). *Search for information* characterizes an active coping style, where patients are looking for social support and information about the disease and treatment options in an active manner (e.g., “I exchanged experiences in dealing with the disease with other patients”). *Search for support in religion* is a scale which reflects the personal preferences for religion as a coping resource (e.g., “I prayed and sought consolation in the faith”). Cronbach’s α in the present study was for *Rumination* 0.73, *Defense of threat* 0.83, *Search for social integration* 0.78, *Search for Information* 0.76, and *Search for support in religion* 0.77.

#### 2.4.4. Emotion Regulation Skills

General emotion-regulation skills were assessed with the Emotion Regulations Skills Questionnaire (ERSQ) [60]. The ERSQ is a 27-item self-report measure that assesses the application of emotion-regulation skills during the previous week on a 5-point Likert scale (1 ‘‘not at all’’ to 5 ‘‘almost always’’). It contains nine scales that correspond to the nine emotion regulation skills. Items are preceded by the stem ‘‘Last week….’’ and include e.g., ‘‘…I paid attention to my feelings’’ or ‘‘…my physical sensations were a good indication of how I was feeling’’. Cronbach’s α in the present study for the total score was 0.95.

#### 2.4.5. Relationship Quality

To assess the statisfaction in the relationship, the Relationship Assessment Scale (RAS) was used [61]. The RAS includes seven items that are rated on a 5-point Likert scale with scores ranging from 1 to 5 (e.g., “Overall, how satisfied are you with your relationship?”). Higher scores reflect higher relationship satisfaction. This questionnaire was only completed by participants with a partner. Cronbach’s α in the present sample was 0.99.

#### 2.4.6. Mental Fatigue

Mental fatigue was assessed by means of an abbreviated 6-item version of the Mental Fatigue Scale (MFS) [62]. Answers were given on a 4-point Likert scale from 0 to 3. Cronbach’s α in the present sample was 0.79.

In addition to the measures above, at the beginning of the therapy, each patient defined her or his individual therapy goals using a modified version of the Goal Attainment Scaling procedure (GAS-R) [63]. Goal attainment scaling involved the following steps: (a) participants were asked to specify the most important therapy goals at baseline; (b) on a continuum of possible outcomes between −2 (worst expected outcome) and +4 (best expected outcome), pretreatment performance was set at 0; (c) criteria for scoring at each level were specified; and (d) participants were asked to evaluate the extent of goal attainment at post-treatment. The therapists were instructed to define at least two goals together with the patients. The average score of all three goals at post-assessment was used as an outcome measure.

## 3. Treatments

As noted above, in the present study, two treatment conditions were compared: the standard treatment (Standard) and the standard plus treatment (Standard PLUS). Both treatment programs were manualized (for a detailed summary, see Table 1). All therapies were carried out by three female psychologists qualified as psychotherapists and neuropsychologists. For every patient, an individual treatment plan was formulated. The duration of the treatment program was not limited a priori, but instead adapted to suit each patient’s individual needs. However, the general aim was to achieve the therapy goals within 20 sessions over the course of one year. 

The Standard treatment contained solely the treatment of neuropsychological disorders caused by an ABI as well as their effects on everyday life. The program was based on conventional recommendations for neuropsychological outpatient rehabilitation [64,65,66]. The psychological interventions followed a mainly psychoeducational rationale focusing on problem-solving and coping. No specific psychotherapeutic interventions as mentioned in the “Standard PLUS” program below, such as focusing on emotional processes, were included.

In the Standard PLUS treatment, neuropsychological interventions were combined with evidence-based psychotherapeutic interventions [46,67]. The psychotherapeutic interventions were adapted from cognitive–behavioral, emotion-focused, process-experiential as well as interpersonal psychotherapies [68,69,70]. The additional psychotherapeutic components included focus on the emotional aspects of coping during the emotional adjustment to the consequences of an ABI, conscious redefinition of one’s role in one’s social and occupational life, and re-evaluating one’s hopes, fears, and expectations for the future.

## 4. Power Analysis

According to power considerations based on previous studies [67,71], we planned to include 72 patients to show a difference of three points on the BDI-II reaching the level of significance (*p* < 0.05). Due to the much slower than anticipated recruitment rate over time, the recruitment had to finish before reaching this target. 

## 5. Statistical Analysis

Group differences in demographics and between-group effects at pre-assessment were analyzed by using chi-square tests and two-tailed *t*-tests. All statistical analyses for the primary outcome and all secondary outcomes were conducted by using intent-to-treat analysis, using the last-value-carried-forward (LOCF) method to account for missing data. Between-group differences at post-treatment and follow-up were analyzed using univariate analyses of covariance (ANCOVAs) using pre-treatment scores as covariate [72]. Within-group changes in each group from pre- to post-treatment and from pre- to follow-up assessment were tested by paired *t*-tests. Hedge’s g was calculated as effect size (ES) for within-group changes. Between-group effect sizes were calculated as the difference of within-effect sizes between the two groups. In addition, the long-term treatment effect for the primary outcome was analyzed using a repeated-measures ANOVA with a time (baseline, post-treatment, and follow-up scores) and a group factor. Pairwise differences were measured using paired *t*-tests with a Bonferroni correction. Because of the relatively small sample size, we also conducted non-parametric tests to complement the parametric tests described above. All analyses were performed in IBM SPSS Statistics version 25.

## 6. Results

### 6.1. Preliminary analyses

Table 2 shows the demographic and injury characteristics of the study sample. There were no significant differences between the two groups regarding age, gender, marital status, employment, neurological disorders, and time since injury at baseline. 

Table 3 shows the descriptives of the questionnaire assessments of both groups at each time point (baseline, post-treatment, and 6-month follow-up). At baseline, there were no differences between the two groups on most of the measures. However, there was a significant difference between the two groups regarding illness awareness from the relatives’ perspective, (t(20) = 2.30, *p* = 0.03; U = 28.0, *p* = 0.03). 

In total, 3 of 25 participants (12%, all ischemic stroke patients) did not complete the post-assessment questionnaires. Dropout analyses revealed no significant differences on all baseline assessments regarding two-tailed t-tests (ps = 0.078– 0.98) as well as non-parametric U-test (ps = 0.12–0.97). All patients, who provided post-treatment data, also completed the follow-up assessment.

The average number of therapy sessions in the Standard PLUS group was 20.6 (SD = 5.0), and 18.3 (SD = 4.1) in the standard group. There was no significant difference between the groups regarding number of sessions that patients received (*t*(23) = −1.27, *p* = 0.21; U = 57.0, *p* = 0.25).

### 6.2. Intervention Effects at Post-Treatment

Analyses of treatment outcome at post-treatment as well as respective within- and between-group effect sizes are reported in Table 4. Regarding the primary outcome, paired t-tests showed a significant within-group decrease in depressive symptoms measured by the BDI-II in both groups. Effect sizes were in the large range. A univariate ANCOVA controlling for pre-treatment depressive symptoms showed no significant difference between the two groups for depressive symptom scores post-treatment.

Regarding secondary outcomes, paired t-tests revealed significant within-group changes in the Standard PLUS group regarding mental fatigue, awareness from all three perspectives, and emotion regulation skills. Significant within-group changes for the standard group were found for mental fatigue, quality of life, acceptance of the disability, rumination, and seeking social support, as well as awareness for the clinician and the patient. Univariate ANCOVAs controlling for pre-treatment scores showed significant differences between the Standard PLUS and Standard group regarding emotion regulation skills measures with the ERSQ at post-treatment, whereas there was no difference between the two groups regarding all the other secondary outcomes at post-treatment. Independent t-tests (t(23) = −3.10, *p* = 0.005) and non-parametric Mann–Whitney U-Test (*p* = 0.004) showed the same results for the ERSQ.

Regarding the average score of all three individualized treatment goals, i.e., GAS, the Standard PLUS (M = 2.47 SD = 1.28) and the Standard group (M = 2.57 SD = 1.29), did not differ significantly from each other (t(20) = 0.17, *p* = 0.86; U = 58.0, *p* = 0.89).

Based on the criteria for reliable change (reduction of pre-treatment BDI-II scores of at least 10 points [73]), 8 patients out of 13 (62%) of the Standard PLUS condition and 4 patients out of 12 (33%) of the Standard group were classified as having achieved a reliable change from pre- to post-treatment. However, statistically, the groups did not differ significantly regarding the number of patients reaching a reliable change χ2 (1, n =25) = 1.98, *p* = 0.16.

### 6.3. Intervention Effects at Follow-Up Assessment

Results regarding the follow-up assessment are described in Table 5. Regarding the primary outcome, paired t-tests demonstrated significant within-group changes on depressive scores for the Standard PLUS as well as for the standard group.

Univariate ANCOVAs controlling for pre-treatment scores showed no significant differences between the two groups on any outcome measure at follow-up assessment. Independent t-tests (ps = 0.11–0.99) and non-parametric Mann–Whitney tests (ps = 0.07–0.99) for follow-up scores showed the same pattern.

Within- and between-group effect sizes at follow-up are presented in Table 5. From baseline to follow-up assessment, large (ES > 0.80) within-group effect sizes were found for the Standard PLUS group on BDI-II and mental fatigue and moderate (ES = 0.50–0.79) within-group effects were found on acceptance of disability and awareness of relatives. Large within-group effect sizes (ES > 0.80) for pre- to follow-up-treatment were found for the standard group on depressive symptoms, quality of life, and emotion regulation, moderate within-group effects (ES = 0.59–0.79) were found on mental fatigue, rumination, seeking social support, and awareness from the patient and the relative perspective.

Finally, a repeated-measures ANOVA of time (pre-treatment, post-treatment, follow-up) x group (Standard PLUS, Standard) for the primary outcome was conducted. Mauchly’s tests indicated that the assumption of sphericity had been violated for the main effect time, χ2(2) = 19.83, *p* < 0.001. Therefore, degrees of freedom were corrected using Greenhouse–Geisser estimates of sphericity (ε = 0.63 for the main effect time). Repeated-measures ANOVA resulted in a significant main effect of time, F(1.25, 28.85) = 26.52, *p* < 0.001, while the interaction time x group was not significant, F(1.25, 28.88) = 0.56, *p* = 0.49. The effect sizes for the main effect of time was ηp² = 0.722 and for the interaction, time x group was ηp² = 0.044.

## 7. Discussion

An acquired brain injury is a drastic incident that has major influences on affected people and their social network. Thus, such events are often accompanied by far-reaching adjustments to be made that go along with mood impairments. The present study compares two individual psychological treatments for patients diagnosed with an AjD after an ABI in a randomized-controlled trial (RCT), i.e., an integrated cognitive–behavioral treatment combined with a neuropsychological therapy (Standard PLUS) vs. a neuropsychological treatment alone (Standard). We hypothesized that the integrated treatment is more effective regarding reduction of depressive symptoms than the standard neuropsychological treatment alone. The results did not support this hypothesis. Whereas both treatments showed large effects on depressive symptoms at post-treatment and at 6-month follow-up, there was no significant difference between the two treatments. The large within-group effect size from baseline to post-treatment for the Standard PLUS condition was in the same range as the effect of the same treatment in an uncontrolled pilot study [46].

The present results regarding depressive symptoms are in line with studies that have compared two active treatments in an RCT in major depression [74]. However, the within-group effect sizes were large and comparable with a meta-analysis that reported beneficial results of psychological intervention for patients after an ABI [42]. Nevertheless, the effect sizes were below benchmarks of within-group effect sizes for depressive symptoms in psychotherapy studies for depression [75]. These results demonstrate that the treatment of patients with acquired brain injuries is especially challenging since several impairments (e.g., neuropsychological and emotional deficits) have to be incorporated at the same time in one treatment [76,77]. Moreover, residual depressive symptoms may partly be attributable to acquired neurological and neuropsychological deficits and could hamper the rehabilitation process and outcome [78,79]. In comparison to other studies with ABI patients comparing active treatments, the standard treatment in the present study may have been a too strong comparator that was too similar to the integrative treatment. Modern neuropsychological interventions are not only limited to restitution and compensation of neuropsychological deficits but also incorporate coping with consequences of the injury as well as the implementation of selected cognitive–behavioral interventions [80,81]. The mentioned result is in line with results of other studies in this population which did not find differences between active conditions when compared to a passive control group [82].

Whereas there was a significant between-group difference for emotion regulation skills at post-treatment, this difference disappeared until the 6-month follow-up. For the other secondary outcomes, there were no group differences at post-treatment or follow-up. Importantly, in both groups, no change was observed regarding the relationship quality. One reason could be that both treatments focused mainly on intrapersonal factors and not on interpersonal aspects, and future treatments should also incorporate a stronger focus on interpersonal factors. Since a brain injury has not only a major impact on the patient alone but also on her or his social environment, future studies might need to regularly incorporate significant others into the psychological treatment.

For many patients, depressive symptoms are a major problem after ABI, may persist for a long time after the injury, and often limit rehabilitation outcomes [83]. Post-stroke aftermath, such as enduring cognitive impairment after stroke [84], a persistent loss of energy, or a permanent sick leave from job [21], is illustrative of the complexity of treatment in mood disorders after ABI. Therefore, the prevention of mood disorders should be an important part of the treatment after ABI. A Cochrane review revealed promising results and found a small significant effect for psychotherapy in the prevention of post-stroke depression [85]. Moreover, in a study from Robinson and colleagues [86] comparing selective serotonin reuptake inhibitors (SSRI), problem-solving therapy, and placebo, both SSRIs and problem-solving therapy were found to be superior to placebo in the prevention of mood disorders after ABI. Therefore, it might be important for stroke survivors to receive specific neuropsychological therapy in combination with psychotherapeutic interventions to prevent chronic disorder courses. Notwithstanding, more systematic research is needed to evaluate the long-term effectiveness of prevention therapy for mood disorders after ABI.

The present study has some noteworthy limitations. A first important shortcoming is the small sample size of the present study. Future studies might consider broadening the inclusion criteria in order to reach a sample size large enough to compare two active treatments. A second point is that standard neuropsychological therapy followed a holistic approach [87] that incorporated many CBT components (e.g., behavior activation, coping strategies, operant conditioning strategies, and a rest management program) [65]. A recent published meta-analysis showed that neuropsychological therapy (e.g., computerized cognitive-training) was associated with improvement in depressive symptoms and everyday functioning [88]. Thus, it can be assumed that the neuropsychological treatment alone has had a positive impact on mood. In that sense, it is not possible to differentiate the effects of the combination therapy from the neuropsychological therapy. Lastly, there were no measures of adherence for both treatments. Since therapists, licensed as neuropsychologist and CBT psychotherapist, provided treatments in both conditions, spillover effects may have occurred that ultimately led to similar outcomes in both conditions. Furthermore, the sample in the present study was rather heterogeneous with regard to the ABI. Future studies might consider investigating more homogenous samples. Potential benefits may include the fact that subgroups with regard to corresponding neurological deficits may profit more from specific psychotherapeutic interventions (e.g., emotion-focus interventions). However, more homogenous samples would limit generalizability and further reduce the feasibility of a study. In addition, future studies might consider a shorter post-injuries time point and include patients, which have already reached a certain level of neurological functioning.

In sum, we did not find that an integrated treatment approach for the treatment of patients with an adjustment disorder after an ABI is superior to a standard approach. Nevertheless, both treatments showed medium to large within-group effects sizes on most measures for patients suffering from an AjD after ABI. To the best of our knowledge, this is the first trial investigating the effectiveness of an integrative treatment compared with an active control condition on patients suffering from AjD after ABI.

## Figures and Tables

**Figure 1 jcm-09-01684-f001:**
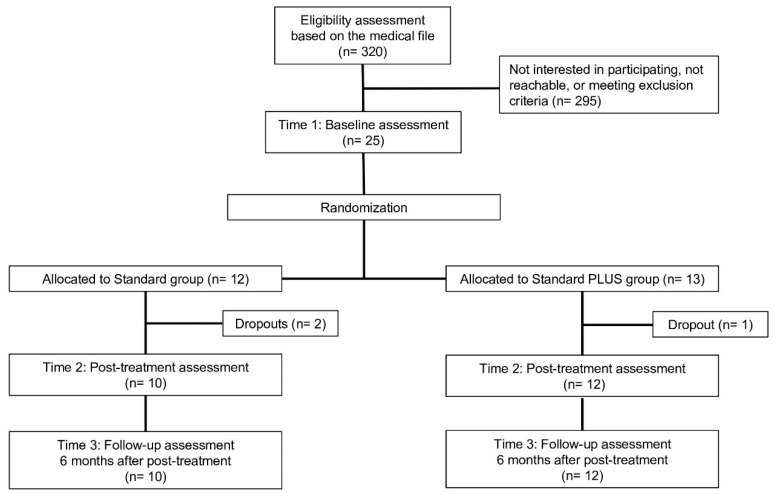
Study flowchart.

**Table 1 jcm-09-01684-t001:** Overview of the treatments.

Phase	Standard Treatment	Standard PLUS Treatment
Preparatory (Week 1)	Establishing a sustainable working alliance, information about the aims and procedures of the therapy. Formulation of individual therapy goals and introduction of a tiredness diary.	Analogs to the standard treatment procedure week 1
Intervention (Week 1–18)	Implementation of the standard therapy, with the aim of achieving individual therapy goals by the 18th session.	Implementation of the standard PLUS Therapy, with the aim of achieving individual therapy goals by the 18th session.
	Important topics: - Improvement of functional level - Achievement of the best possible degree of independence in everyday life - Provision of information: education and information about the brain injury and its consequences ^a^ - Dealing with a changed degree of tiredness	Important topics: - Improvement of functional level - Achievement of the best possible degree of independence in everyday life - Provision of information: education and information about the brain injury and its consequences ^a^ - Dealing with a changed degree of tiredness - Training of emotional competence - Adjustment to the changed possibilities
	Therapeutic interventions: - Restitution ^b^ of cognitive functions. - Compensation ^b^ for any functional deficits (e.g., the use for external memory aids) - Coping ^b^ with the consequence of injury (e.g., memory problems, dealing with tiredness) - Integrated techniques as operant conditioning techniques (e.g., cognitive self-teaching techniques, self-management) ^c^	Therapeutic interventions: - Restitution ^b^ of cognitive functions. - Compensation ^b^ for any functional deficits (e.g., the use for external memory aids) - Coping ^b^ with the primary (e.g., memory problems, dealing with tiredness) and secondary consequence of injury - Integrated techniques as operant conditioning techniques (e.g., cognitive self-teaching techniques, self-management) ^c^
		*PLUS: Psychotherapeutic techniques (e.g., emotional-focus therapy and clarification ^d^, activating resources ^e^, generating, positive expectancies, normalization)* *The principle of mindfulness based on “Mindfulness-Based Cognitive Therapy” ^f^*
Maintenance (Week 18–20)	Not intended as an actual therapy session. Overall review, consolidation of what has been achieved, and reiterating the individual strategies involved. Discussion potentially difficult situations and planning of coping methods.	Analogs to the standard treatment procedure week 18–20.

Notes. Additional components of the Standard PLUS condition are presented in italics. ^a^ Lukens and McFarlane, 2005; ^b^ Gauggel et al., 2003; ^c^ Fuster, 1997; ^d^ Greenberg, 2002; ^e^ Flückiger et al. 2009.; ^f^ Segal, Williams and Teasdale, 2002.

**Table 2 jcm-09-01684-t002:** Sample characteristics for the Standard PLUS and Standard treatment.

	PLUS (n = 13)	Standard (n = 12)	Statistic
Age, years (SD)	50.8 (7.8)	45.6 (12.6)	t(23) = −1.24, *p* = 0.22
Gender, *n* (%)			χ2(1) = 0.05, *p* = 0.82
Male	6 (46.1)	5 (41.6)	
Female	7 (53.9)	7 (58.4)	
Marital status, *n* (%)			χ2(2) = 1.22, *p* = 0.54
Single/living alone	2 (15.4)	4 (33.3)	
Married/living together	9 (69.2)	6 (50.0)	
Divorced	2 (15.4)	2 (16.7)	
Education in years (SD)	12.4 (1.1)	12.6 (2.9)	t(23) = 0.17, *p* = 0.86
Employment, *n* (%)			χ2(2) = 1.10, *p* = 0.56
Self-Employed / Partner	5 (38.4)	6 (50.0)	
Student	1 (7.8)	0	
Sick pay/unfit for work	7 (53.8)	6 (50.0)	
Neurological disorder, *n* (%)			χ2(4) = 3.10, *p* = 0.54
Ischemic stroke	6 (46.1)	9 (75.0)	
Hemorrhagic stroke	3 (23.1)	2 (16.7)	
Traumatic brain Injury	2 (15.4)	1 (8.3)	
Tumor	1 (7.7)	0 (0.0)	
Encephalitis	1 (7.7)	0 (0.0)	
Time since injury, months (SD)	14.1 (13.4)	19.6 (25.7)	*t*(23) = 0.67, *p* = 0.50

**Table 3 jcm-09-01684-t003:** Descriptives of outcome measures over time.

	Baseline					Post		Follow-up	
Measure	M	SD	Between-Group ^1^			M	SD	M	SD
			df	T ^a^	U ^b^				
BDI-II			23	−1.07	58.00				
PLUS	21.77	9.56				11.15	5.13	10.92	4.11
Standard	17.92	8.24				10.50	5.50	8.00	4.59
MFS			22	−0.81	57.00				
PLUS	9.92	2.67				7.50	2.68	7.00	2.82
Standard	8.83	3.74				6.58	2.74	6.91	3.31
WHOQol			23	−1.02	54.00				
PLUS	91.00	12.75				95.53	13.08	96.77	11.00
Standard	86.00	11.42				94.41	9.34	95.16	10.33
ADS			23	−0.31	72.00				
PLUS	78.15	8.07				73.38	10.94	72.83	8.70
Standard	76.92	11.34				72.50	9.42	72.84	10.63
TSK_RU			23	1.08	61.00				
PLUS	25.69	7.79				25.23	7.99	23.15	7.38
Standard	28.83	7.38				23.83	5.65	23.91	6.51
TSK_SS			23	−0.41	71.50				
PLUS	36.00	7.38				38.07	4.87	38.69	4.99
Standard	34.83	6.49				38.33	6.28	38.08	5.93
TSK_BA			23	−1.21	59.50				
PLUS	37.08	5.48				36.38	4.85	36.23	6.71
Standard	33.75	8.09				35.50	6.72	34.25	7.24
TSK_SI			23	0.57	71.00				
PLUS	25.77	6.07				25.61	6.55	24.38	7.38
Standard	27.33	7.63				28.58	6.93	26.33	8.56
TSK_SR			23	−0.28	71.50				
PLUS	8.54	4.01				8.15	3.60	7.92	4.55
Standard	8.08	4.23				8.83	3.29	7.33	2.87
AQ_R			20	2.30 *	28.00 *				
PLUS	33.92	5.01				40.08	5.90	38.25	6.09
Standard	39.20	5.75				42.30	6.03	42.20	5.78
AQ_C			23	−1.39	54.50				
PLUS	46.00	3.13				50.23	3.32	------	------
Standard	43.75	4.86				48.16	6.13	------	------
AQ_P			23	−0.08	72.50				
PLUS	37.77	5.70				40.92	6.73	39.61	6.64
Standard	37.58	6.24				40.75	5.92	40.91	6.18
RAS			20	−1.19	44.50				
PLUS	25.17	2.16				24.91	1.83	26.79	4.26
Standard	23.90	2.80				25.50	3.34	24.90	2.88
ERSQ			23	−1.18	58.00				
PLUS	70.92	14.54				85.00	13.69	78.46	13.72
Standard	52.50	20.80				66.91	15.29	71.00	17.29

Notes. BDI-II = Beck Depression Inventory—Second Edition; MFS = Mental Fatigue Scale; AQ_R = Awareness Questionnaire Relative; AQ_C = Awareness Questionnaire Clinician; AQ_P = Awareness Questionnaire Patient; RAS = Relationship Assessment Scale; ERSQ = Emotion Regulations Skills Questionnaire; WHOQOL = World Health Organization Quality of Life; ADS = Acceptance Disability Scale; TSK = Trier Coping with Disease Scales; TSK_RU = Rumination; TSK_SS = Seeking social support, TSK_BA = Avoidance, TSK_SI = Information seeking, TSK_SR = Seeking help in religion. ^1^ Between-groups differences at baseline using *t*-test ^a^ and U-test ^b^, * *p* < 0.05.

**Table 4 jcm-09-01684-t004:** Results of analyses of covariance at post-treatment (T2) controlled for baseline (T1) and within- and between-group effect sizes at post-treatment.

Measure	Group	Mean Difference (T1-T2)	95% CI	Within-Group			Between-Group		
				df	T	Effect Size	df	F	Effect Size ^1^
BDI-II	PLUS	10.70	(2.01–19.38)	12	3.53 **	1.32	1,22	<0.01	0.30
	Standard	7.33	(1.51–13.14)	11	3.51 **	1.02			
MFS	PLUS	2.90	(0.51–5.29)	11	2.59 *	0.87	1,21	0.15	0.22
	Standard	2.33	(0.84–3.82)	11	4.18 **	0.65			
WHOQol	PLUS	−6.20	(−13.76–1.36)	12	−1.61	−0.34	1,22	0.35	−0.43
	Standard	−7.78	(−13.87–−1.67)	11	−3.38 **	−0.77			
ADS	PLUS	6.10	(−1.97–14.17)	12	1.64	0.48	1,22	<0.01	0.07
	Standard	3.11	(−1.95–8.17)	11	2.41 *	0.41			
TSK_RU	PLUS	2.01	(−2.68–6.68)	12	0.22	0.06	1,22	2.19	−0.68
	Standard	4.89	(1.90–7.87)	11	4.10 **	0.74			
TSK_SS	PLUS	−3.00	(−7.66–1.66)	12	−1.23	−0.32	1,22	0.24	−0.21
	Standard	−2.89	(−6.44–0.66)	11	−2.63 *	−0.53			
TSK_BA	PLUS	1.20	(−2.56–4.96)	12	0.53	0.13	1,22	0.22	0.10
	Standard	−2.89	(−8.11–2.33)	11	−0.96	−0.23			
TSK_SI	PLUS	−0.10	(−6.00–5.80)	12	0.07	0.02	1,22	0.84	−0.14
	Standard	−2.22	(−7.91–3.47)	11	−0.65	−0.16			
TSK_SR	PLUS	0.40	(−0.99–1.79)	12	0.77	0.08	1,22	1.18	−0.10
	Standard	−1.33	(−3.67–1.01)	11	−0.79	−0.18			
AQ_R	PLUS	−6.40	(−9.65–−3.14)	11	−4.86 ***	−1.09	1,19	0.56	0.59
	Standard	−3.67	(−7.43–0.09)	9	−1.97	−0.50			
AQ_C	PLUS	−4.80	(−7.27–−2.32)	12	−4.25 ***	−1.30	1,22	0.03	0.51
	Standard	−4.11	(−7.07–−1.14)	11	−3.92 **	−0.79			
AQ_P	PLUS	−2.50	(−5.22–0.22)	12	−2.91 *	−0.50	1,22	<0.01	0.01
	Standard	−3.22	(−6.74–0.29)	11	−2.74 *	−0.49			
RAS	PLUS	0.00	(−1.01–1.01)	11	0.61	0.14	1,19	2.23	−0.36
	Standard	−1.78	(−4.14–0.58	9	−1.71	−0.50			
ERSQ	PLUS	−14.50	(−28.40–−0.59)	12	−2.84 *	−0.97	1,22	7.56 *	0.21
	Standard	−9.89	(−20.82–1.04)	11	−0.93	−0.76			

Notes. BDI-II = Beck Depression Inventory—Second Edition; MFS = Mental Fatigue Scale; AQ_R = Awareness Questionnaire Relative; AQ_C = Awareness Questionnaire Clinician; AQ_P = Awareness Questionnaire Patient; RAS = Relationship Assessment Scale; ERSQ = Emotion Regulations Skills Questionnaire; WHOQOL = World Health Organization Quality of Life; ADS = Acceptance Disability Scale; TSK = Trier Coping with Disease Scales; TSK_RU = Rumination; TSK_SS = Seeking social support, TSK_BA = Avoidance, TSK_SI = Information seeking, TSK_SR = Seeking help in religion. ^1^ Delta of within-group effect sizes at post-assessment. * *p* < 0.05; ** *p* < 0.01; *** *p* < 0.001.

**Table 5 jcm-09-01684-t005:** Results of analyses of covariance at follow-up (T3) controlled for baseline (T1) and within- and between-group effect sizes at follow-up.

Measure	Group	Mean Difference (T1-T2)	95% CI	Within-Group			Between-Group		
				df	T	Effect Size	df	F	Effect Size ^1^
BDI-II	PLUS	10.30	(1.76–18.83)	12	3.65 **	1.42	1,22	1.91	−0.02
	Standard	10.11	(5.06–15.15)	11	5.46 ***	1.44			
MFS	PLUS	3.30	(1.06–5.53)	11	3.41 **	1.02	1,21	0.32	0.50
	Standard	1.78	(−0.24–3.80)	11	2.73 *	0.52			
WHOQol	PLUS	−8.20	(−16.46–0.06)	12	−1.83	−0.46	1,22	0.35	−0.34
	Standard	−9.55	(−17.65–−1.45)	11	−3.17 **	−0.80			
ADS	PLUS	6.70	(−1.22–14.62)	12	1.90	0.62	1,22	0.05	0.26
	Standard	3.67	(−0.94–8.28)	11	2.12	0.36			
TSK_RU	PLUS	2.54	(2.95–6.87)	12	0.22	0.32	1,22	2.19	−0.36
	Standard	4.91	(−2.00–7.07)	11	4.11 **	0.68			
TSK_SS	PLUS	−2.72	(−6.39–1.01)	12	−1.23	−0.42	1,22	0.24	−0.09
	Standard	−3.33	(−6.75–0.25)	11	−2.63 *	−0.51			
TSK_BA	PLUS	−0.81	(−2.73–4.42)	12	0.53	0.14	1,22	0.22	0.06
	Standard	−0.50	(−5.48–4.48)	11	−0.96	−0.08			
TSK_SI	PLUS	1.43	(−4.01–6.78)	12	0.07	0.19	1,22	0.84	0.07
	Standard	1.01	(−4.51–6.51)	11	−0.65	0.12			
TSK_SR	PLUS	0.67	(−0.74–1.97)	12	0.77	0.13	1,22	1.18	−0.08
	Standard	0.81	(−0.96–2.46)	11	−0.79	0.21			
AQ_R	PLUS	−5.10	(−7.65–−2.54)	11	−3.76 **	−0.76	1,19	0.08	0.26
	Standard	−3.22	(−6.29–−0.15)	9	−1.15	−0.50			
AQ_P	PLUS	−1.60	(−4.97–1.77)	12	−1.52	−0.30	1,22	0.73	−0.21
	Standard	−2.67	(−6.29–0.95)	11	−2.74 *	−0.51			
RAS	PLUS	−0.40	(−2.15–1.35)	11	−0.64	−0.42	1,19	0.03	0.08
	Standard	−1.11	(−3.33–1.11)	9	−1.15	−0.34			
ERSQ	PLUS	−7.30	(−23.23–8.63)	12	−1.39	−0.51	1,22	0.52	−0.42
	Standard	−13.89	(−24.86–−2.91)	11	−1.87	−0.93			

Notes. BDI-II = Beck Depression Inventory—Second Edition; PHQ-9, Patient Health Questionnaire; Self-Assessment Questionnaire MFS = Mental Fatigue Scale; AQ_R = Awareness Questionnaire Relative; AQ_C = Awareness Questionnaire Clinician; AQ_P = Awareness Questionnaire Patient; RAS = Relationship Assessment Scale; ERSQ = Emotion Regulations Skills Questionnaire; WHOQOL = World Health Organization Quality of Life; ADS = Acceptance Disability Scale; TSK = Trier Coping with disease Scales; TSK_RU = Rumination; TSK_SS = Seeking social support, TSK_BA = Avoidance, TSK_SI = Information seeking, TSK_SR = Seeking help in religion. ^1^ Delta within group at follow-up-assessment. * *p* < 0.05; ** *p* < 0.01; *** *p* < 0.001.

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
