# Peer review of "An Integrative Neuro-Psychotherapy Treatment to Foster the Adjustment in Acquired Brain Injury Patients—A Randomized Controlled Study"

_jcm, 2020, doi:10.3390/jcm9061684_

Round 1
Reviewer 1 Report
I agree that AjD is an issue that has not been adequately researched despite its large negative impact on rehabilitation outcomes and quality of life post brain injury. However the specific rational as to how psychological and psychotherapeutic interventions help as stand alone and as integrated combined treatments.
The small number of patients actually included (only 25 out of 320 eligible patients) is a limitation to the study (mentioned in lines 451-3) but this should be expanded further - e.g., why were strict inclusion criteria taken?. Perhaps a different statistical approach could be taken and patients from both groups be matched with each other into pairs according to demographic and clinical criteria - then allowing within-subject statistics (the subject being a patient-pair, one patient from each group).
Minor comments
(1) Please change "familie systematic therapy" to "family and systemic therapies" (line 123).
(2) Please change "sufficient" to "insufficient" (line 157).
(3) Please change Figure 1 to higher resolution.
(4) Please change "assessesapplication" to "assesses the application" (line 229).
(5) Please change "weekon" to "week on" (line 229).
(6) Please change "at posttreatment" to "post treatment (line 248).
(7) Please change "found" to "find" (line 426).
(8) Please change "an" to "a" (line 436).
Reviewer 2 Report
The present study is a randomized controlled trial comparing Standard neuropsychological therapy to Standard PLUS therapy (emotion focus and mindfulness) to reduce adjustment disorder after acquired brain injury. With a primary outcome of reduction in BDI-II scores, the study is very interesting but unfortunately appears to lack power to detect differences between groups. The manuscript was, as a general comment, a little distracting to read with various formatting and font styles throughout the manuscript.
My main consideration is that this ought to be described more as a pilot study, if authors were only able to recruit ~33% of their target population. As the authors also point out, secondary concerns are the similarity of Standard and Standard PLUS treatments.
If word limits allow, authors should provide more detail in the methods and discussion to describe the differences between their therapies, the benefits/drawbacks of homogeneous ABI samples and differing times post-injury, and why larger studies might see differences.
Minor comments:
- Stroke should be mentioned in the abstract as the ABI of interest.
- Throughout the manuscript (but particularly abstract and discussion) authors should explain their timepoints as they have elsewhere: baseline, post-treatment and 6 month follow-up. "post" alone is a little confusing.
-
Typo – ‘trails’ instead of ‘trials’ line 87.
-
Timeframe is a bit confusing with regard to comparison of text in abstract – if my understanding is correct, then participants are recruited at any time after 6 months, have a baseline test, and have another test 6 months after. The way the abstract and parts of methods read, they are testing initially after stoke. This should be clarified throughout.
-
Stoke is included as an inclusion criteria, but then authors go on to say they widened their criteria to include TBI and encephalitis. This should be clearer.
-
2 sections ‘recruitment of patients’ and ‘patients’ should be combined into one.
-
It is unclear from Figure 1 whether “control” is the Standard treatment group? If so, should be labelled this way.
-
It would be interesting to the reader to know which 3 were dropouts – were they all stroke, or other ABIs?
-
It should be clarified in text that all participants who went through treatment were followed up at post-treatment and 6m timepoints (i.e. no dropouts after the 6 months?)
- Figure 1 is of low image quality - it might just be the resolution in the PDF version, but a higher resolution figure should be provided if possible.
Author Response
Please see the attachment

This manuscript is a resubmission of an earlier submission. The following is a list of the peer review reports and author responses from that submission.